# Research on the Jobs-Housing Balance of Residents in Peri-Urbanization Areas in China: A Case Study of Zoucheng County

**Haonan Zhang** , **Hu Zhao \*** , **Saisai Meng** and **Yanghua Zhang**

School of Architecture and Urban Planning, Shandong Jianzhu University, Jinan 250100, China;
2020055117@stu.sdjzu.edu.cn (H.Z.); 2021050112@stu.sdjzu.edu.cn (S.M.); zhangyh@mail.bnu.edu.cn (Y.Z.)
\* Correspondence: zhaohu@sdjzu.edu.cn

**Abstract:** In the process of urbanization, peri-urbanization is a unique phenomenon in China. For residents living in peri-urbanization areas, realizing the balance between workplace and living space is not only a crucial guarantee for them to secure livelihood but also an important criterion to measure the quality of China's urbanization. Based on the questionnaire data distributed by the research group in Zoucheng County, China, in 2021, this study measures the degree of jobs-housing balance in county area by constructing the benefit index of jobs-housing balance and explores factors affecting the jobs-housing balance in county area by using logistic regression, random forest classification, and regression tree. Results: Firstly, with 57% of the residents have achieved the standard, the level of jobs-housing balance in Zoucheng County is relatively high. Secondly, individual, household and built environment dimensions jointly affect jobs-housing balance of residents. Furthermore, at the current stage of China's county areas, household is not the core dimension influencing jobs-housing balance. Thirdly, factors that passed the significance test can be divided into three categories: key factors, important factors, and auxiliary factors. Occupation type, commuting way and residential location are the key factors affecting the jobs-housing balance, which deserve our attention. Therefore, according to the above conclusions, relevant suggestions for promoting jobs-housing balance of residents in county area were put forward. For instance, considering the diverse occupation of local residents, employments that match their skills should be offered, and as for peri-urbanization areas, the regulatory of jobs-housing balance should be placed in urban and rural areas.

**Keywords:** benefit; peri-urbanization; county area; jobs-housing balance; analysis of influencing factors



## 1. Introduction

Since the 21st century, China's urbanization level has begun to catch up and even surpass the global average. According to the World Bank, from 2000 to 2020, the global urbanization rate increased from 46.7% to 56.16%, an increase of 9.46% in 20 years. Moroever, a report released by the Department of Economic and Social Affairs of the United Nations, found that the urban population will continue to grow, reaching 6 billion in 2041 [1]. Additionally, by 2041, China's urbanization rate will increase by 27.67% from 36.22% to 63.89%, with a three-times growth rate of the world average. Joseph E. Stiglitz, the Nobel laureate in economics, said that the high-tech development in the United States and urbanization in China will be two key influencing factors in the development of human society in the 21st century [2]. The rapid urbanization process has brought about great social changes in China. In particular, there is a large number of rural migrants in cities and towns [3]. According to the National Bureau of Statistics of China, from 2000 to 2020, about 445 million additional rural migrants in cities and towns. To them, obtaining a decent job and a comfortable house is almost everyone's ideal [4]. However, it is difficult for these rural migrants to have satisfactory housing and jobs at the same time. Moreover, a large number of rural migrants have also had an adverse impact on the living and working condition of the indigenous

people in county towns. Therefore, jobs-housing balance, which is the coordination level between rural migrants' workplace and living space, is an important evaluation dimension of regional urbanization quality. Furthermore, the analysis of factors affecting jobs-housing balance is of great significance for the realization of high-quality urbanization.

On the one hand, urbanization has promoted the socioeconomic development of all countries in the world and has greatly raised people's quality of life. On the other hand, a phenomenon that cannot be ignored is peri-urbanization in the process of urbanization. On the global scale, there was always a clear boundary between a city and its surrounding environment until the 20th century; thus, it was suitable to use two different concepts of city and village to describe the form of spatial development at that time [5]. However, since the beginning of the 21st century, due to political, economic, social, environmental, and other various reasons, the differences between a city and its surrounding areas cannot be clearly distinguished as in the past [6]. Under the current situation, metropolises have lost their traditional boundaries due to dispersion and disorderly expansion. As they move towards the existing living space, agricultural land, and natural environment, a peri-urbanization phenomenon has been formed [7]. Peri-urbanized areas, far from the metropolitan boundary and close to rural areas, are generally in rural-urban fringe zone, among which county area is a good representative area [8]. The phenomenon of peri-urbanization has not only attracted worldwide attention but is prominent in China with certain particularity. Due to the restrictions of China's unique hukou (household registration) system, in a wide range of county areas in China, there are not only the above-mentioned manifestations of peri-urbanization, but also a large number of "peri-urbanization" population, and it can be said that the county is a representative area of peri-urbanization [9]. As the emergence of peri-urbanization, it has a negative impact on residents living in county towns, especially the jobs-housing relationship of rural migrants. First of all, due to the high cost of living, a large number of rural migrants are employed in county towns but live in villages, which results in a large number of urban-rural commutes. Long-time and long-distance commuting not only increases the commuting cost of rural migrants but also brings adverse effects on their physical and mental health. Furthermore, for residents in peri-urbanization areas, although most of them have realized individual urbanization, their families are still stuck in the old way, causing social problems such as left-behind children, women, and the elderly. The last point, as the high cost of housing, a large number of migrant workers may choose to rent in counties, but it will make them have a poor sense of belonging, vulnerable to discrimination by indigenous people, and thus induce violent crime.

This paper focuses on the county area because of the following peculiarities. First of all, from the perspective of social attributes, counties have been the basic unit of China's national governance for more than two thousand years, whereas the geographic names and patterns of many counties have survived to this day [10]. Moreover, as one of the administrative regions in China, counties play a connecting role, implementing the work of the central government, provinces, and municipalities, leading towns and communities, promoting socio-economic development, and serving the people's livelihood. In addition, from the perspective of economic attributes, the development of the county's economy is an important carrier for coping with the international financial crisis and developing the domestic economic cycle [11]. According to 2020 China's Top 100 County Economy Research, China's top 100 counties account for less than 2% of the country's land and 7 % of the country's population, whereas their GDP is about one-tenth of the whole country, indicating that the county economy has become a new drive of high-quality economic development. Last but not the least, from the perspective of population attributes, county area is pivotal for China to promote urbanization and gain citizenship for rural migrants. According to the data from the National Bureau of Statistics of China, in 2020, the permanent residents of county towns in China was 226,932,700, accounting for about 25.21% of the country's urban permanent population, and compared with 2010, the permanent residents

of county towns increased by 614,469,000, with an average annual growth rate of about 3.7% [4].

Consequently, it is necessary to measure the jobs-housing balance of non-agricultural employees in county areas, to analyze the factors influencing the process of population urbanization in peri-urbanization areas of China. The innovations of this article are as follows: On the one hand, the New Economics of Labor Migration (NELM) holds that families adhere to the principle of division of labor among household members when deciding to migrate, maximizing expected income while minimizing household economic risks [12]. Therefore, this paper takes household attribute as a core influencing factor to analyze individual jobs-housing balance. On the other hand, in previous studies, the criteria for judging whether individuals have achieved jobs-housing balance were mainly based on cost-orientation, that is, only based on commuting time and commuting distance [13,14]. However, in this paper, the measurement of jobs-housing balance not only considers the cost but also the benefit. Therefore, this paper proposes a new indicator to measure the jobs-housing balance, which is the benefit index of jobs-housing balance.

To sum up, this study set out to construct a new measurement indicator of jobs-housing balance, namely, the benefit index of jobs-housing balance, and to explore the influencing mechanism of jobs-housing balance in the county area. The overall structure of the study is as follows: Firstly, the cost-to-income ratio of the financial field is used to analyze the jobs-housing balance, whereas the benefit index of jobs-housing balance is constructed to judge whether an individual achieves jobs-housing balance. Depending on it, 18 influencing factors are selected from individual, household, and built environment dimensions to develop a binary logistic regression model, to further analyze these influencing factors. In addition, random forest classification and regression tree are used to measure the degree of the influencing factors that passed the significant test in binary logistic regression. This paper enriches the theoretical connotation of jobs-housing balance in theory and provides a reference for the government to regulate residents' jobs-housing space and realize jobs-housing balance in practice. The structure of the rest of this paper is as follows: The Section 2 reviews the related studies from three aspects, including the urbanization of county areas in China, the criteria for judging jobs-housing balance, and the influencing factors of jobs-housing balance. The Section 3 is concerned with the research area, data, methods, and variables used for this study. The Section 4 presents the results of the model. The Section 5 is the discussion, and the conclusion is in the Section 6.

## 2. Literature Review

### 2.1. Urbanization in China's County Area: The Neglected Peri-Urbanization Areas

From the perspective of the spatial relationship of jobs-housing, the existing urbanization types of China's population can be divided into two categories considering hukou: trans-regional urbanization and nearby urbanization. Trans-regional urbanization mainly refers to the spatial large-scale, long-distance, and population separating from the registered residence of non-agricultural employment groups in central and western China or economically underdeveloped areas to the developed areas in the east [15,16]. It has been the mainstream of China's urbanization since the reform and opening up, but in this urbanization model, it is difficult for migrant workers to achieve the jobs-housing balance, mainly because of the high housing costs and complicated settlement policies of eastern Chinese cities [17]. As a result, a large number of non-agricultural workers from the central and western regions or economically backward areas are unstable in the eastern part of China, and there has even been a clear trend of return in recent years, which means that some migrant workers have returned to the county areas. Therefore, the necessity of promoting another type of urbanization is exposed, that is, nearby urbanization. As a type of urbanization actively promoted by the national new urbanization strategy in recent years, nearby urbanization is opposed to the urbanization away from home and is different from in situ urbanization and rural urbanization. In the process of nearby urbanization, the spatial direction of urbanization of the rural population is mainly county towns, that is

sub-districts and town areas in the prefecture and counties, whereas the county area is the front line of the transformation of registration and economic income distribution [18,19]. Therefore, nearby urbanization can also be considered as the urbanization of county area. Under the guidance of China's new urbanization strategy, the county area will become a key carrier of rural migrants, whereas county area urbanization should be an important trend in future urbanization. However, the peri-urbanization of county areas is a major phenomenon of county urbanization that should arouse people's attention [20,21]. From the perspective of population and geography, peri-urbanization has two meanings in China [22]. The first means peri-urbanized population. Although the occupation type and workplace of rural migrants have changed, it is tough for them to live or even settle down in cities, due to the high cost of living and the strict hukou system. In addition, even if they have the opportunity to work and live in the city at the same time, their citizen status is only "as it should be", and there is no "real" citizen power so they rarely can enjoy the welfare and public services of the city, which seriously affects the jobs-housing relationship. The second is the peri-urbanized area, that is, the area near or between large cities (including county areas), which are closely related to large cities, and their labor-intensive industries are developing rapidly. Moreover, in terms of population and geography, most of China's counties are peri-urbanized. Given all that, county area urbanization is the focus of China's future urbanization, and the phenomenon of peri-urbanization, as an important urban issue in the process of county urbanization, needs to be solved urgently, which not only has a significant impact on jobs-housing relationship and jobs-housing spatial structure for residents in county area but also more profoundly affects the process of high-quality urbanization in China. Therefore, priority should be given to the above issue.

## 2.2. Criteria for Judging the Jobs-Housing Balance

There are many definitions of jobs-housing balance, which can be explained from two perspectives. At a macro level, it is mainly determined by judging whether an area's jobs-housing ratio matches, that is, the ratio of the number of employed people to the number of households in a geographical unit. Cervero [23] believed that assuming that only one person per household in a study area is employed and each household has its independent housing; thus, the area is absolutely balanced when the jobs-housing ratio is equal to 1; the area lacks sufficient housing to meet the local labor force when the ratio is higher than 1; the area is also in an imbalance when the ratio is less than 1. It is widely used in the related research of jobs-housing balance. For example, Xu [24] used the jobs-housing ratio as an indicator to measure the jobs-housing balance when he studied the impact of the jobs-housing balance on traffic safety in Los Angeles, and the ratio between 0.75 and 1.5 was considered an effective jobs-housing balance [25]. Zhou [26] and Huang [27] also used the same ratio to study the impact of jobs-housing balance on urban space. As an indicator of urban planning, the jobs-housing ratio is usually used for comprehensive analysis and policy formulation in specific areas. However, the balance between the number of workers and the number of houses is not exactly equivalent to the actual jobs-housing balance. Giuliano [28] found out that between the 1970s and 1990s, population growth, the increasing use of cars, the increasing proportion of female workers, and the insufficient construction of expressways all contributed significantly to traffic congestion, which was irrelevant to the jobs-housing spatial distribution. Based on the study of Taipei City, Chen [29] found that the jobs-housing balance had little impact on commuting, and questioned the hypothesis of cost minimization. In addition, the second is the jobs-housing balance based on the perspective of micro individuals, which mainly includes the measurement methods based on distance and time. Commuting distance and time can reflect the level of jobs-housing balance of individuals in a more comprehensive and real-time manner. In terms of commuting distance, Zhang [30] and Zhou [31] used commuting distance as a measurement index to study the jobs-housing balance in specific areas. Based on data from two censuses, Li [32] studied the change in commuting patterns in Australia, especially the change in commuting distance. In terms of commuting time,

extensive work has been done by scholars in the area of using bus card data to track individuals' homes and workplaces, and then calculating the commuting time to judge jobs-housing balance of groups [33,34]. Excess commuting based on the cost of commuting time is also a heated research topic of jobs-housing balance [35–37]. Regarding the standard of commuting time and distance, it is generally believed that if the commuting time is within 30 min [38,39], the commuting distance is within 5 km [40,41], the individual is in a comfortable commuting state, that is, the jobs-housing balance has been achieved. However, there are certain limitations in the measurement criteria of jobs-housing balance in the above study, as only considering commuting time and commuting distance cannot fully reflect the individual's jobs-housing balance status. Therefore, the benefit index of jobs-housing balance was proposed for further research.

### 2.3. Influencing Factors of Jobs-Housing Balance

A reasonable jobs-housing relationship is conducive to reducing commuting volume, urban traffic congestion, and commuting time, as well as improving people's experience of going out and their life quality. Paying attention to jobs-housing balance in a city is essential to the quality of urban development. Therefore, many studies have researched influencing factors of the jobs-housing balance. After a numerous literature collection and collation, this paper divides the influencing factors of jobs-housing balance into two aspects, namely individual factors and built environment factors. As the subject of realizing jobs-housing balance, the individual's own attributes cannot be ignored; thus, individual factors of jobs-housing balance can be analyzed from three angles, including population attributes, social attributes, and economy attributes. In terms of population attributes, Dai [38] took the middle class in Guangzhou as the research object and found out that gender and age are significant factors that affect the jobs-housing balance. In terms of social attributes, Watts [42] used the comprehensive spatial econometric model to study the determinants of the average commuting distance between the workplace and the housing, and concluded that education level has a more significant impact on jobs-housing balance. Li [43] explored the influencing factors of the jobs–housing separation in Beijing and found that due to the relaxation of the hukou system, the role of hukou on the jobs-housing separation has changed from negative to positive. In terms of economic attributes, the focus is on the individual income. By analyzing the commuting distance sensitivity of different working groups, Ding [40] found that the commuting threshold of workers with different incomes is various, which further affects the jobs-housing balance. Among them, the commuting threshold is the distance at which workers are willing to commute to work. The second major influencing factor of the jobs-housing balance is the built environment, which can be divided into three categories, namely, living environment, employment environment, and public service environment. From the perspective of living environment, Huang [33] discussed the commuting characteristics of groups with different living patterns. From the perspective of employment environment, Zhou [44] analyzed the employment accessibility between different commuting ways; Murphy [35] found in his study of excess commuting in Dublin, Ireland, that commuting using private transportation is more efficient and conducive to jobs-housing balance; Zhu [45] found the influence of eight different commuting ways on commuting happiness, and put forward relevant measures to improve jobs-housing balance. In terms of public services environment, Djurhuus [46] studied the relationship between public transportation and self-reported active commuting, and found that distance to bus stops, density of bus stops, and modes of transportation were all positively correlated with active commuting through multi-level logistic regression. As China enters the stage of a new type of urbanization, people are aching for permanent settlement with their household, rather than temporary settlement of individuals, whereas they also have a desire to enjoy the same rights and treatment as people with urban hukou. Therefore, household is increasingly becoming an important influencing factor in the jobs-housing balance for individuals. However, far too little attention has been paid to the household factor, and even if some of the studies involved household-related factors,

they were fragmentary. For example, Neto's [47] study of commuting time in the São Paulo metropolitan area of Brazil demonstrated that marital status has a greater impact on commuting times of working women. Using data from the 2017 U.S. National Household Travel Survey, Hu [48] analyzed differences in commuting distance between five different household types. In this paper, however, the household factor is innovatively regarded as a first-level indicator, and it is integrated into the system of influencing factors of jobs-housing balance from the perspective of household structure and different generational roles, to study the effect of household on individual jobs-housing balance.

*2.4. Research Framework*

All this leads up to the theoretical framework of this study, as shown in Figure 1. First of all, in previous studies, the measure of individual jobs-housing balance was based on cost orientation, that is, commuting time and commuting distance. According to the cost-to-income ratio, the benefit index of jobs-housing balance was proposed, which is efficiency-oriented, that is, both the commuting cost and the income of residents were considered. In addition, the influencing factors of jobs-housing balance are analyzed from three aspects, namely, individual, built environment, and household. According to the previous review, it can be found that the individual factor and the built environment factor have been more common, whereas there are few studies analyzing the impact of the household factor on jobs-housing balance. In the context of China's new urbanization, a large number of rural immigrant families have been separated based on "economic rationality" in the past, resulting in many social problems [49], therefore, household should be considered as an important dimension in the model. Finally, based on the three-factor coupling of individuals, built environment, and household, the significant influence factors and the intensity of them are explored, and the mechanism affecting residents' jobs-housing balance is summarized from the two dimensions of the category and intensity of the influencing factors.

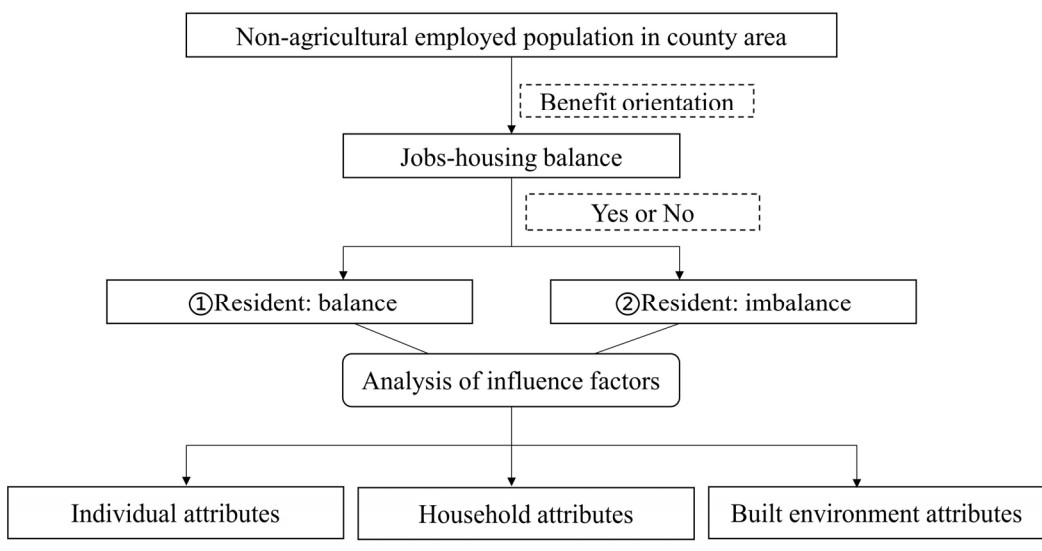

**Figure 1.** Research framework.

## 3. Research Design

### 3.1. Research Area

Zoucheng County (116°44′30″~117°28′54″ E, 35°09′12″~35°32′54″ N) is located in the west-south of Shandong Province and the southeast of Jining City, as shown in Figure 2. By 2020, Zoucheng has jurisdiction over 3 sub-districts and 13 towns, with an administrative area of 1616 square kilometers, a permanent resident population of 1,166,600, and a regional GDP of 82.412 billion yuan. With a long history and less population outflow, the urbanization of Zoucheng County is mainly local urbanization, which can be confirmed by

the data of the seventh population census in China. In 2020, the permanent population of Zoucheng County was 1,166,559, with a total increase of 49,867 in ten years, an increase of 4.47%. Among the permanent residents in the county, 755,941 people live in cities and towns, accounting for 64.80%; 410,618 people live in rural areas, accounting for 35.20%. Compared with 2010, the urban population increased by 242,523, the rural population decreased by 192,656, and the proportion of urban population increased by 18.82% [50]. In addition, the urbanization rate of Zoucheng was 64.8 % in 2020, whereas the average level of China's top 100 counties was 64.9%, which means that Zoucheng County is a typical example of county areas urbanization in China.

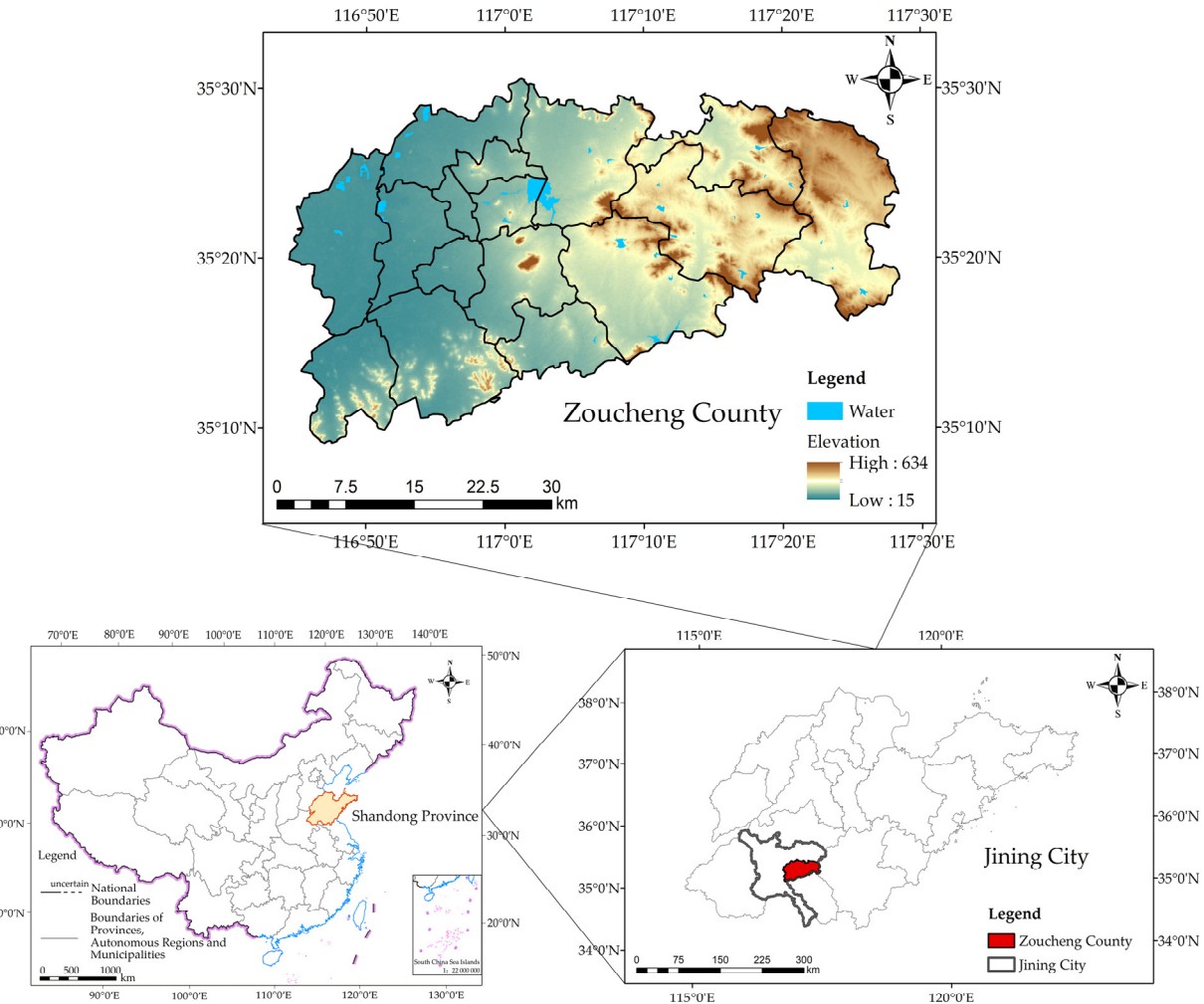

**Figure 2.** Research area.

Moreover, Zoucheng County has certain representativeness among cities at the same level in Shandong Province, China. This study analyzes the economic and population development of 26 county-level cities in Shandong Province in 2020, with GDP per capita in terms of economy and population density in terms of population. (Data were obtained from the official website of the county's bureau of statistics.) The result shows that Zoucheng County is at a medium level of development, ranking 11th in GDP per capita and 13th in population density, as shown in Figure 3. Although the development of it is moderate, Zoucheng's urbanization rate is relatively high, reaching 65%, which is superior to the average level in China, indicating that Zoucheng is well represented in this study.

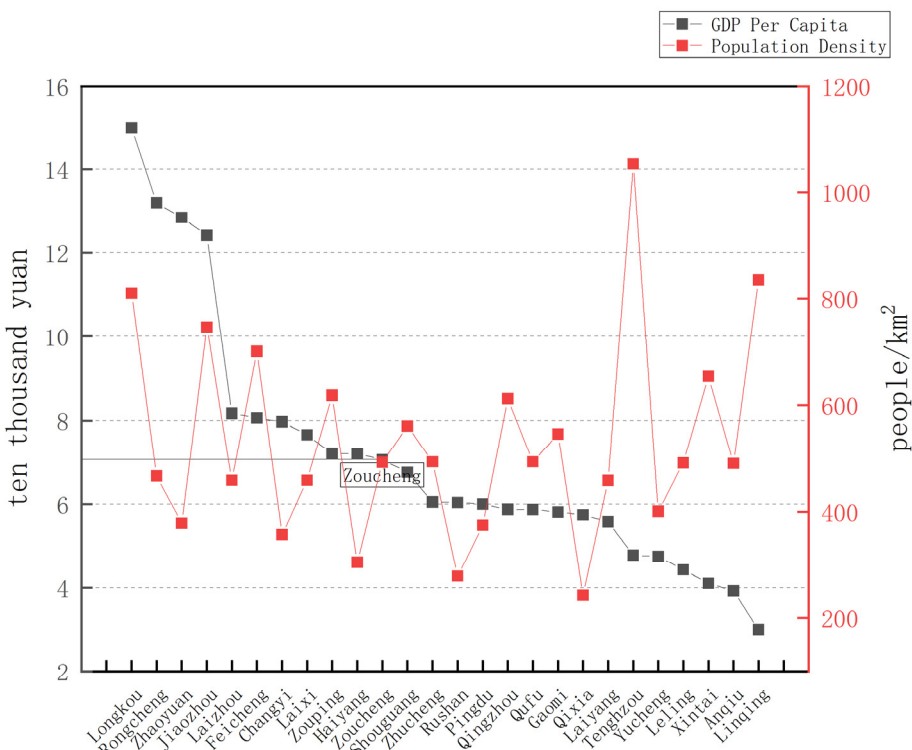

**Figure 3.** GDP per capita and population density of 26 county-level cities in Shandong Province.

*3.2. Data*

3.2.1. Questionnaire Data Collection

The data for this study are mainly from the household questionnaire (Supplementary Material) survey conducted in Zoucheng County from 17–30 November 2021. It should be stressed that the COVID-19 in Zoucheng County was relatively stable, and there were no new confirmed cases during the survey. Please see *Jining Municipal Health Commission* (http://wjw.jining.gov.cn/col/col59856/index.html, accessed on 15 June 2022). Therefore, the closed-off management was not implemented by the government, nor did it affect the travel patterns of interviewees. Since the purpose of this paper is to study the influencing factors of non-agricultural employees' jobs-housing balance in the process of county urbanization, there are 35 questions from three aspects in the questionnaire, including individual, household, and built environment. PPS sampling survey method was used to ensure the representativeness of the sample, that is, several villages or communities were randomly selected from 3 sub-districts and 13 towns throughout Zoucheng County, and questionnaires were distributed in proportion according to the economic development level and population distribution of each sub-district and town. In addition, professional investigators are randomly assigned to conduct household surveys, and each household selects a non-agricultural practitioner as a representative, to ensure that there is no gender bias in the research, the respondents can be heads of households or non-heads of households. (Due to China's hukou system, men are generally the head of the household.) In this way, the respondents are allowed to fill out the questionnaire face to face. With database management via SPSS software in this study, 1110 valid samples were finally screened after data cleaning and the elimination of invalid samples. The sample distribution was shown in Figure 4. To be clear, the *Communique on Major Data of the Seventh National Census of Zoucheng County* showed that the permanent residents' population was 1,166,559 in 2020. The sample size of this study is 1110, and the sampling ratio is about 1000:1. It concluded that the sample size complied with the statistical requirements with the 99% of confidence coefficient and the 4 of confidence interval.

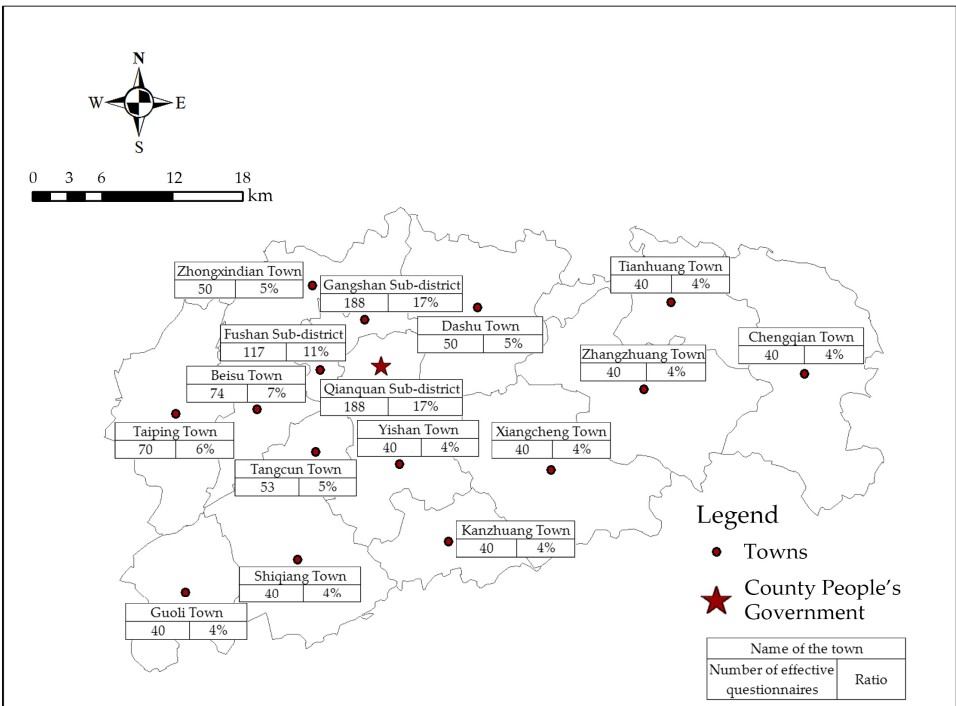

**Figure 4.** Spatial distribution map of questionnaires.

### 3.2.2. Basic Information of Respondents

(1)　Economic and social characteristics of respondents

The following is a descriptive analysis of the basic attributes and jobs-housing status of 1110 samples obtained from the survey. The basic attributes of the sample are mainly described from six aspects, namely age, gender, education level, income, hukou, and occupation type, as shown in Table 1. In terms of age, the age distribution of the survey group is mainly concentrated in 20 to 50 years old, that is, most of them belong to young adults, which are in the golden stage of the labor force; thus, the sample is representative, reflecting jobs-housing status of most non-agricultural employees in Zoucheng County. In terms of gender, the ratio of man to woman is 3:2, which is consistent with the gender distribution in the Zoucheng County Statistical Yearbook. In terms of education level, the academic qualifications of non-agricultural employees in Zoucheng County are mostly high school and below, which is also in line with the talent situation of the county. As the living conditions and employment opportunities in counties are far less than those in large cities, highly educated talents tend to choose large cities, whereas people with lower academic qualifications have poor survivability in large cities, and in order to balance nostalgia and pursue modern life, they are more likely to stay in the county, and proved by the hukou status. (Hukou status refers to whether local residents have a local household registration. It is a household-based population management system implemented by the government of the People's Republic of China for its citizens who settle down on the mainland of China. The legality of natural persons' behavior of living and working in a certain place can be determined by the system). In this county, 96% of the workers are registered, and the migrant population is small. In terms of occupation types, the largest number of self-employed individuals reached 39%, indicating that the tertiary industry in Zoucheng County is relatively developed. Moreover, Zoucheng County has the largest per capita monthly income at the level of 2001 to 3500 yuan, reaching 31%, which is still far from the minimum average monthly wage standard of 4166 yuan for the Chinese middle class.

**Table 1.** Social and economic characteristics of respondents.

| Age | Rate% | Gender | Rate% | Education Level | Rate% |
|---|---|---|---|---|---|
| ≤20 | 2 | man | 60 | Junior high school and below | 46 |
| 21~30 | 20 | woman | 40 | High School | 29 |
| 31~40 | 37 | | | JuniorCollege | 12 |
| 41~50 | 22 | | | Bachelor | 12 |
| >50 | 19 | | | Master's degree and above | 1 |
| **Income (RMB)** | **Rate%** | **Hukou** | **Rate%** | **Occupation types** | **Rate%** |
| ≤2000 | 17 | registered in local urban areas | 30 | employees of government agencies and public institutions | 16 |
| 2001~3500 | 31 | registered in local rural areas | 66 | enterprise employees | 10 |
| 3501~5000 | 28 | registered in urban areas from other counties | 2 | Workers | 21 |
| 5001~8000 | 16 | registered in rural areas from other counties | 2 | social service personnel | 14 |
| ≥8000 | 8 | | | Self-employed individuals | 39 |

(2)　Jobs-housing status of the respondents

On the whole, Zoucheng County has a sound jobs-housing condition, characterized by less commuting time, shorter commuting distance, and more than half of people choosing green travel modes. Specifically, 89% of individuals commute within 30 min, and 66% commute within 5 km. Among all commuting ways, the proportion of green travel modes such as walk, bicycle, electromobile, bus, and shuttle bus is as high as 70%. See Figure 5.

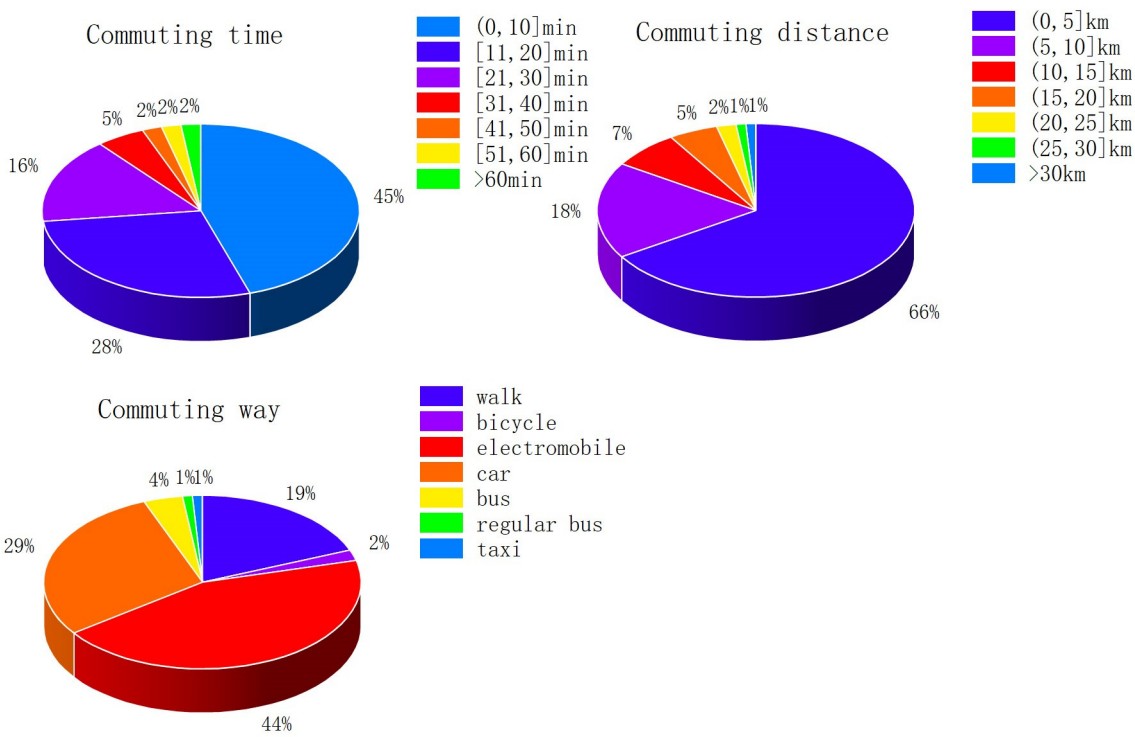

**Figure 5.** Jobs-housing status of non-agricultural employees in Zoucheng County.

*3.3. Research Methods*

3.3.1. Benefit Index of Jobs-Housing Balance

In this paper, the benefit index of jobs-housing balance, which is based on the cost-to-income ratio in the financial fields [51,52], is used as the criterion to judge whether non-agricultural employees achieve jobs-housing balance. The cost-to-income ratio is the ratio of the bank's operating expenses to its operating income, reflecting how much a

bank spends on each unit of income. Additionally, it is an important indicator to measure the profitability of the bank. Referring to the cost-to-income ratio, the benefit index of jobs-housing balance is constructed:

$$JBI = \frac{I}{0.5CT + 0.5CD} \tag{1}$$

In this formula, the benefit is represented by income, and recorded as *I*. Costs are expressed in terms of commute time plus commute distance, which are recorded as *CT* and *CD*, respectively. In order to make different variables comparable, income, commuting time and commuting distance are divided into five levels according to the Likert scale, with each level corresponds to a score, as shown in Figure 6. The benefit index of jobs-housing balance is the ratio of income to commuting cost, reflecting the income corresponding to each unit of commuting cost, that is, the higher the ratio is, the higher the degree of work-housing balance is. In this study, the judgment standard of jobs-housing balance is: if *JBI* is greater than or equal to 1, residents have achieved jobs-housing balance; if *JBI* is less than 1, the opposite is true.

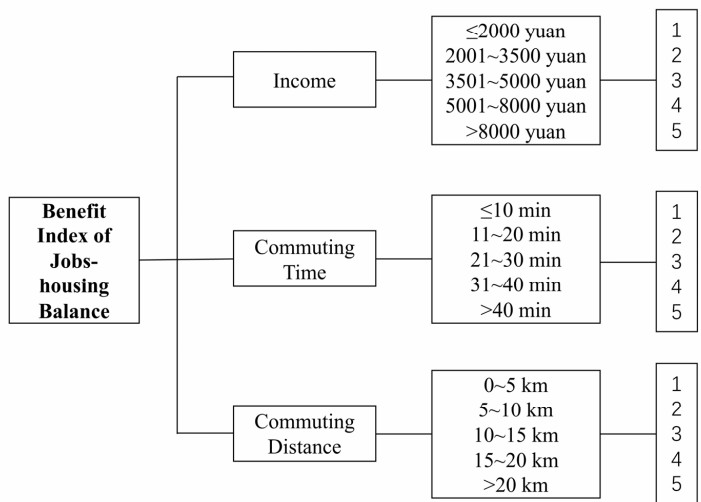

**Figure 6.** Composition of benefit index of jobs-housing balance.

### 3.3.2. Logistic Regression

Logistic regression is commonly used when the dependent variable is a binary variable [53]. According to the purpose of the study, the dependent variable in this paper is whether the urban and rural non-agricultural employees have achieved the jobs-housing balance, and the value of 1 is considered to be realized, whereas the value of 0 is regarded as unrealized. Considering the dependent variable in this paper is a binary variable, the binary logistic regression model can be used to analyze the influencing factors of the jobs-housing balance for non-agricultural employees in urban and rural areas. Assuming that there is a linear relationship between dependent variable *y* and independent variable $x_k$, then:

$$y = \beta_0 + \beta_1 x_1 + \beta_2 x_2 + \cdots\cdots \beta_k x_k + \varepsilon \tag{2}$$

If *P* is used to represent the probability of an event under a given condition, that is, the probability of *y* = 1, the relationship between the conditional probability of an event and the independent variable $x_k$ is obviously not linear:

$$P_i = \frac{1}{1 + e^{-(\alpha + \sum_{i=1}^{n} \beta_i * x_i)}} \tag{3}$$

$$1 - P_i = 1 - \frac{1}{1 + e^{-(\alpha + \sum_{i=1}^{n} \beta_i * x_i)}} \tag{4}$$

In the formula, $P_i$ represents the probability of the event occurring in the $i$th observation, and $1 - P_i$ represents the probability of the event not occurring in the $i$th observation. Moreover, the Ratio of event occurrence probability $P_i$ to non-occurrence probability $1 - P_i$, $P_i / 1 - P_i$ is called the event Odds Ratio. By logarithmic transformation of Odds Ratio, a linear logistic regression model can be obtained:

$$\ln\left(\frac{P}{1-P}\right) = \alpha + \beta_1 x_1 + \beta_2 x_2 + \cdots\cdots + \beta_n x_n + \varepsilon \tag{5}$$

In the formula: $P$ is the probability of the occurrence of the set event, which refers to whether non-agricultural employees can achieve jobs-housing balance, that is, the probability of occurrence of "jobs-housing balance = 1". Moreover, $x_1, x_2, \ldots, x_n$ is the explanatory variable, and n is the number of independent variables; $\alpha$ is a constant term, $\beta_1, \beta_2, \ldots, \beta_n$ is partial regression coefficient of logistic regression, and $\varepsilon$ is random error. In addition, $\beta_n$ is positive, indicating that the $n$th factor has a positive impact on the jobs-housing balance for non-agricultural employees; $\beta_n$ is negative, indicating that the $n$th factor has a negative impact on the jobs-housing balance for non-agricultural employees.

The prediction ability of Logistic regression model is evaluated by obtaining maximum likelihood estimation, which includes regression coefficient, index Exp(B) of parameter estimation, as well as standard deviation, Wald statistics, and significance level estimated by regression coefficient.

### 3.3.3. Random Forest Classification and Regression Tree

Random Forest Classification and Regression Tree (CART) uses Gini impurity for feature selection. Gini impurity, also known as Gini coefficient, reflects the probability that two samples randomly selected from dataset $D$ but belong to different categories. If every data item in the set belongs to the same classification, the error rate is 0, and the lower the Gini impurity value, the better the reaction classification effect and the higher the order degree of the set. As for the CART, assuming that the sample set $D$ has $k$ classifications, and the probability that a sample belongs to the $k$th classification is $p_k$, the purity of the data set $D$ can be measured by Gini coefficient.

$$Gini(D) = \sum_{k=1}^{k} p_k(1 - p_k) = 1 - \sum_{k=1}^{k} p_k^2 \tag{6}$$

Assuming that the sample set $D$ has multiple characteristics, select one of the characteristic values $a$ and assign it to the variable $A$, then $A = a$ is obtained. Based on the CART, classification trees divides $D$ into two parts, that is, $D_1$ is the sample set that meets $A = a$, and $D_2$ is the sample set that does not satisfy the feature. Then, the Gini coefficient of sample set $D$ is expressed as follows under the condition of $A = a$:

$$Gini(D, A) = \frac{|D_1|}{|D|} Gini(D_1) + \frac{|D_2|}{|D|} Gini(D_2) \tag{7}$$

$Gini\ (D, A)$ represents the uncertainty of set $D$ after the division of $A = a$. Moreover, the CART starts from the root node and recursively builds $D$ classification trees with $D$ training sets. After, calculate the Gini coefficients of each existing feature of the current node, select feature $A$ with the smallest Gini coefficient as the optimal feature, and take the corresponding value a as the optimal segmentation point. Each CART decision tree in *a* random forest continuously divides the dataset into two subsets by constantly traversing the feature subset of the tree, looking for the feature segmentation point with the smallest Gini coefficient, until the stop condition is met.

### 3.4. Variable Selection

#### 3.4.1. Independent Variables

Independent variables are divided into three categories, namely, individual factors, household factors, and built environment factors, as show in Table 2. In this paper, individual factors include gender, age, education level, hukou, and occupation type. The whole household migration can be regarded as the key condition for the permanent change of migration, and a series of decisions made by individuals are ultimately aimed at the interests of the whole household. Therefore, regarding the influencing factors of jobs-housing balance, priority is given to the influence of household. Household factors include household structure, marital status, taking children to school, the number of children, and education stage of their children. Built environment factors include living environment, employment environment, and public service environment. The living environment is measured by the residential location, the willingness to relocation, housing tenure, and the housing area; the employment environment is measured by the type of workplace and the commuting way; the public service environment is measured by the walking time to the nearest bus stop and the satisfaction of service facilities near the workplace.

**Table 2.** Variables used in this study.

| Variables | Variable Code | Value | Value Meaning | Sample Distribution | Variable Attribute | Variable Code | Value | Value Meaning | Sample Distribution |
|---|---|---|---|---|---|---|---|---|---|
| Individual factor | $X_1$ | 1 | ≤20 years old | 1.80% | | $X_9$ | 1 | one child | 58.8% |
| | | 2 | 21–30 years old | 20.27% | | | 2 | two children | 41.2% |
| | | 3 | 31–40 years old | 37.30% | | $X_{10}$ | 1 | Children | 4.2% |
| | | 4 | 41–50 years old | 22.16% | | | 2 | Children's grandparents or maternal grandparents | 13.5% |
| | | 5 | >50 years old | 18.47% | | | 3 | Parents | 33.2% |
| | $X_2$ | 1 | man | 60.00% | | | 4 | other | 49.1% |
| | | 2 | woman | 40.00% | | $X_{11}$ | 1 | village | 24.68% |
| | $X_3$ | 1 | junior high school and below | 45.86% | | | 2 | town | 31.17% |
| | | 2 | high school | 28.65% | Built environment factors | | 3 | sub-district | 44.14% |
| | | 3 | junior college | 12.16% | | $X_{12}$ | 1 | yes | 84.22% |
| | | 4 | bachelor | 11.89% | | | 2 | no | 15.78% |
| | | 5 | master degree and above | 1.44% | | $X_{13}$ | 1 | village | 1.90% |
| Y | $X_4$ | 1 | urban hukou | 87.60% | | | 2 | town | 56.90% |
| | | 2 | rural hukou | 11.20% | | | 3 | sub-district | 41.20% |
| | $X_5$ | 1 | employees of government agencies and public institutions | 16.49% | | $X_{14}$ | 1 | private transportation | 94.68% |
| | | 2 | enterprise employees | 10.45% | | | 2 | public transportation | 5.32% |
| | | 3 | worker | 20.81% | | $X_{15}$ | 1 | ≤5 min | 60.51% |
| | | 4 | social service workers | 13.60% | | | 2 | 6–10 min | 30.25% |
| | | 5 | self-employed individuals | 38.65% | | | 3 | 11–15 min | 5.34% |
| Household factor | $X_6$ | 1 | couple family | 14.14% | | | 4 | 16–20 min | 1.54% |
| | | 2 | stem family | 13.06% | | | 5 | >20 min | 2.36% |
| | | 3 | single family | 12.07% | | $X_{16}$ | 1 | yes | 90.80% |
| | | 4 | nuclear family | 60.72% | | | 2 | No | 9.20% |
| | $X_7$ | 1 | married | 88.8% | | $X_{17}$ | 1 | house with property right | 87.66% |
| | | 2 | single | 11.2% | | | 2 | house without property right | 12.34% |
| | $X_8$ | 1 | not enrolled to school | 45.77% | | $X_{18}$ | 1 | ≤60 m² | 4.70% |
| | | 2 | preschool | 19.91% | | | 2 | 61–90 m² | 13.30% |
| | | 3 | primary school | 18.29% | | | 3 | 91–120 m² | 45.60% |
| | | 4 | Junior high school | 5.77% | | | 4 | 121–150 m² | 27.60% |
| | | 5 | high school | 10.27% | | | 5 | >150 m² | 8.80% |

#### 3.4.2. Dependent Variables

Considering the influencing factors of the jobs-housing balance of non-agricultural employees are worth studying, the individual non-agricultural employees are selected

as a basic analysis unit. As the dependent variable in this paper is binary, according to the calculation of the benefit index of jobs-housing balance, if the respondent achieves jobs-housing balance, the value is equal to 1; otherwise, it is 0.

## 4. Results

According to the calculation of the benefit index of jobs-housing balance, the proportion of people who have achieved jobs-housing balance in Zoucheng County is 57%.

### 4.1. Logistic Regression Model

In this section, IBM SPSS Statistics 26 software is used to construct the binary Logistic regression model of influencing factors of jobs-housing balance for non-agricultural employees in Zoucheng. The variance inflation factor of all variables is less than 10, which indicates that there is no multicollinearity problem in the model. After calculation, the relevant diagnostic parameters are: Cox and Snell R Square and Nagelkerke R Square of the model are high, reaching 53.2% and 61.3%, respectively, and the prediction accuracy of the model is as high as 70.8%. In addition, the overall significance of the model is 0.951, and it has passed the Hosmer Lemeshow test. Detailed results are shown in Table 3.

**Table 3.** Parameter estimation results of logistic regression model.

| | | B | S.E. | Wald | Sig. | Exp(B) | 95% C.I. for Exp(B) | |
| --- | --- | --- | --- | --- | --- | --- | --- | --- |
| | | | | | | | Lower | Upper |
| | constant | 0.534 | 1.023 | 0.272 | 0.602 | 1.705 | | |
| Gender | woman | −0.503 | 0.151 | 11.095 | 0.001 *** | 0.605 | 0.450 | 0.813 |
| Occupation type | Employees of government agencies and public institution | | | 48.212 | 0.000 *** | | | |
| | enterprise employees | 0.462 | 0.301 | 2.351 | 0.125 | 1.587 | 0.879 | 2.863 |
| | worker | 0.224 | 0.296 | 0.571 | 0.450 | 1.251 | 0.700 | 2.237 |
| | social personnel worker | 0.284 | 0.311 | 0.837 | 0.360 | 1.329 | 0.723 | 2.443 |
| | self-employed individuals | 1.350 | 0.283 | 22.841 | 0.000 *** | 3.858 | 2.218 | 6.713 |
| Household structure | couple family | | | 6.053 | 0.100 * | | | |
| | stem family | −0.732 | 0.301 | 5.900 | 0.015 ** | 0.481 | 0.267 | 0.868 |
| | single family | −0.483 | 0.584 | 0.682 | 0.409 | 0.617 | 0.196 | 10.940 |
| | nuclear family | −0.458 | 0.244 | 3.525 | 0.060 * | 0.633 | 0.392 | 10.020 |
| Residential locations | village | | | 26.786 | 0.000 *** | | | |
| | town | 1.075 | 0.208 | 26.699 | 0.000 *** | 20.931 | 10.949 | 40.408 |
| | sub-district | 0.438 | 0.234 | 3.500 | 0.061 * | 10.550 | 0.979 | 20.452 |
| Commuting way | Public traffic | −2.307 | 0.496 | 21.601 | 0.000 *** | 0.100 | 0.038 | 00.263 |
| Walking time to the nearest bus stop | ≤5 min | | | 15.750 | 0.003 *** | | | |
| | 6~10 min | −0.447 | 0.158 | 7.982 | 0.005 *** | 0.639 | 0.469 | 0.872 |
| | 11~15 min | −0.233 | 0.318 | 0.536 | 0.464 | 0.792 | 0.424 | 1.479 |
| | 16~20 min | −1.461 | 0.599 | 5.941 | 0.015 ** | 0.232 | 0.072 | 0.751 |
| | >20 min | −1.053 | 0.519 | 4.111 | 0.043 ** | 0.349 | 0.126 | 0.965 |
| Housing tenure | House without property right | 0.761 | 0.260 | 8.538 | 0.003 *** | 2.140 | 1.285 | 3.564 |

Note: To simplify, only the factors that passed the significance test in binary logistic regression analysis are listed in the table. ***, **, and * represent significance at the 1%, 5%, and 10% levels, respectively.

### 4.2. Classification Analysis of Influencing Factors

4.2.1. Influence of Individual Factors on Residents' Jobs-Housing Balance

As 5 influencing factors were selected at the individual level, two of them were found to be significant through binary logistic regression analysis, namely gender and occupation type. See Figure 7 for the type, *p*-value, OR value, and 95% CI of the factors that have passed the significance test. First of all, gender has a significant impact on the jobs-housing balance, with the *p*-value of 0.001 and OR value of 0.605 for woman, indicating that in this experiment, taking man as the reference group, the possibility of woman achieving jobs-housing balance is 0.605 times than that of man, whereas the degree of jobs-housing balance of man is higher than that of woman. Secondly, the self-employed individuals in the occupation type significantly affect the jobs-housing balance, with the *p*-value of 0.000 and OR value of 3.858, which means that in this experiment, taking employees of government agencies and public institutions as the reference group, the possibility of self-employed individual achieving the jobs-housing balance is 3.858 times than that of them. However,

enterprise employees, workers, and social service personnel in the occupation type have no significant influence on the jobs-housing balance.

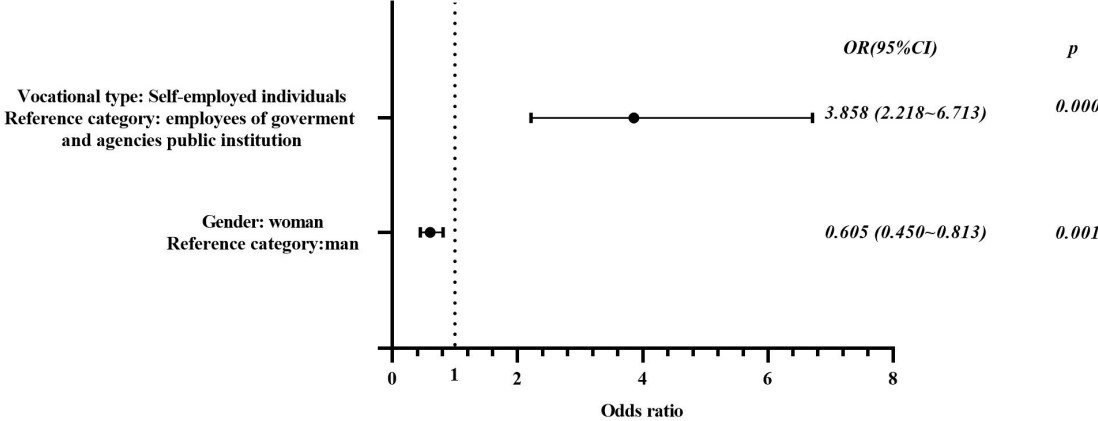

**Figure 7.** OR diagram of individual factors.

### 4.2.2. The Influence of Household Factors on Residents' Jobs-housing Balance

As 5 influencing factors were selected at the household level, only one was found to be significant through binary logistic regression analysis, namely household structure. See Figure 8 for the type, *p*-value, OR value, and 95% CI of the factors that passed the significance test. The stem family and nuclear family in the family structure have a significant influence on the jobs-housing balance. For the stem family, the *p*-value is 0.015 and the OR value is 0.481, which shows that in this experiment, taking the couple family as the reference group, the possibility of the stem family achieving the jobs-housing balance is 0.481 times than that of the couple family, and the degree of the jobs-housing balance of the couple family is higher than that of the stem family. The *p*-value of nuclear family is 0.060, and the OR value is 0.633, indicating that the possibility of nuclear family achieving jobs-housing balance is 0.633 times than that of couple family as the reference group. Therefore, the jobs-housing balance of couple family is the highest, followed by the nuclear family, and finally the stem family. All these show that the more generations of household members, the worse the jobs-housing balance.

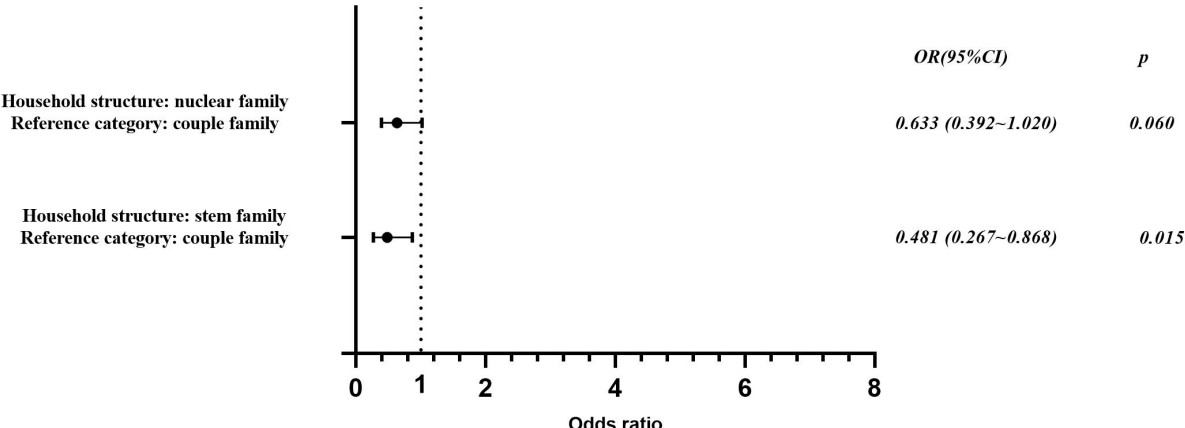

**Figure 8.** OR diagram of household factors.

### 4.2.3. The Influence of Built Environmental Factors on Residents' Jobs-Housing Balance

As 8 influencing factors were selected at the built environment level, four of them were found to be significant through binary logistic regression analysis, namely residential location, commuting way, walking time to the nearest bus stop, and housing tenure. See Figure 9 for the type, *p*-value, OR value, and 95%CI of the factors that passed the significance

test. First of all, the residential location has a significant effect on the balance of work and housing. The *p*-value of people living in towns is 0.000, and the OR value is 2.931, which indicates that in this experiment, taking people living in villages as the reference, the possibility of commuters living in towns achieving jobs-housing balance is 2.931 times than that of commuters living in villages; The *p*-value and OR value of people living in urban areas are 0.061 and 1.550, respectively, indicating that the possibility of commuters living in urban areas to achieve jobs-housing balance is 1.550 times than that of those living in villages. To sum up, the jobs-housing balance of residents living in the town is the best, followed by urban areas, and finally villages. Therefore, it can be seen that the jobs-housing balance shows an inverted U-shaped trend with the improvement of living conditions. Secondly, commuting way has a significant impact on the jobs-housing balance. With the *p*-value of 0.000 and OR value of 0.100 for people who take public traffic, and taking people who take private transportation as a reference, the possibility of people who take public traffic to achieve the jobs-housing balance is 0.1 times than that of the reference group, whereas the degree of jobs-housing balance of people who take private transportation is better. Thirdly, the walking time to the nearest bus stop also has a significant impact on the jobs-housing balance. The *p*-value of the people who walk to the nearest bus stop for 6–10 min is 0.005, and the OR value is 0.639. This shows that people who walk to the nearest bus stop for 6–10 min are 0.639 times more likely to achieve the jobs-housing balance than those who walk to the nearest bus stop for 5 min. The *p*-value and OR value of people who walk for 16~20 min are 0.015 and 0.232, which means that people who walk within 16–20 min are 0.232 times more likely to achieve jobs-housing balance than those who walk within 5 min. People who walk for more than 20 min have a P value of 0.043 and an OR value of 0.349, which means that people who walk to the nearest bus stop for more than 20 min are 0.349 times more likely to achieve jobs-housing balance than those who walk for less than 5 min. In conclusion, the relationship between the walking time to the nearest bus stop and the jobs-housing balance is already obvious, and as the time of walking to the nearest bus stop becomes longer, the possibility of achieving the jobs-housing balance is first reduced and then increased, showing an obvious U-shaped curve. Fourthly, whether or not to own the house with property rights is also one of the important factors that affect the jobs-housing balance. The *p*-value and OR value of the group without property rights are 0.003 and 1.285, respectively, indicating that taking the group with property rights as the reference, the possibility of the group without property rights achieving jobs-housing balance is 1.285 times that of the group with property rights. Hence, it further shows that the jobs-housing balance degree of the non-property property commuters is high.

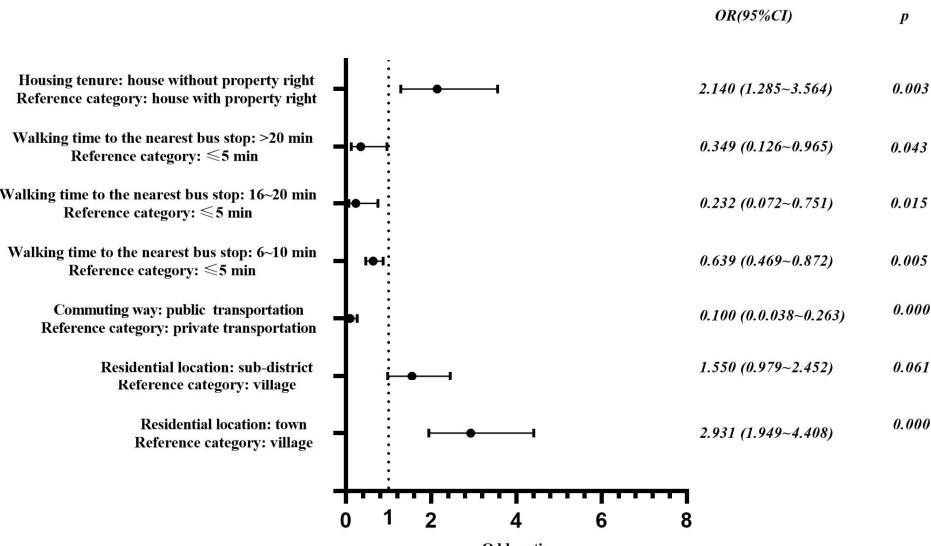

**Figure 9.** OR diagram of built environmental factors.

### 4.3. Intensity Analysis of Influencing Factors

In this section, seven significant factors obtained by binary logistic regression in the previous parts are analyzed by the CART model for determining the influencing intensity. The intensity scores of each factor were obtained through random forest classification and regression tree, as shown in Figure 10.

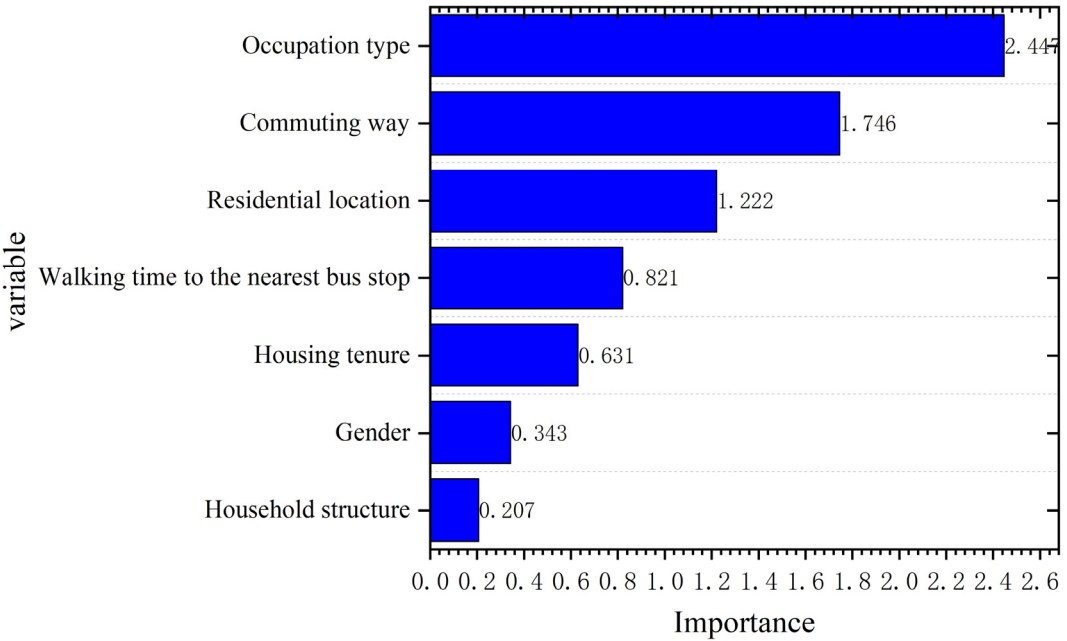

**Figure 10.** Intensity score of significance factors.

Among all the factors that affect the jobs-housing balance, the highest score is the occupation type, which reaches 2.447, indicating that the occupation type has the most significant influence on the jobs-housing balance. Followed by commuting way (1.746), residential location (1.222), walking time to the nearest bus stop (0.821), housing tenure (0.631), and gender (0.343). The lowest score is household structure, only 0.207, which has the least influence on the jobs-housing balance.

## 5. Discussion

### 5.1. The Individual-Household-Built Environment Jointly Affects the Jobs-Housing Balance of Residents in Peri-Urbanized Areas in China

First and foremost, 57% of the residents in Zoucheng County have achieved the jobs-housing balance. Zhang [30] used big data to study the jobs-housing balance in Shanghai and found that at least 57% of residents can work in their respective districts to achieve the jobs-housing balance. By using the travel diary data of Chicago Metropolitan in 2007, and examining two different geographical scales, which are "locality" defined by population census and the integrated construction "central community", Merlin [54] found that 47.6% of the people in the central community have the jobs-housing balance, whereas that of the locality is 33.3%. Compared with the above cities, it is then concluded that there is a relatively high level of jobs-housing balance in Zoucheng County.

Secondly, the factors that passed the significance test of binary logistic regression are analyzed from three aspects, including individual, household, and built environment. At the individual level, gender and occupation type are the influencing factors of jobs-housing balance. From the perspective of gender, men have a sound jobs-housing balance. Although, Crane [55] believed that gender is the main influencing factor of the jobs-housing balance in his research about America, he said that there is little difference in commuting between man and woman, which has a little difference from the drawn conclusion of this paper. This may be because, with the development of the women's liberation movement in the

United States, women are more involved in the economy; therefore, families will consider the commuting of woman workers when choosing a place of residence, whereas under the profound influence of the traditional philosophy of Confucianism, men are responsible for earning money to support their families and women are responsible for housework in China, especially in county areas. As a result of this, the commuting needs of men who are the main labor force of a family will be met first, and consequently, man have a sound jobs-housing balance. From the perspective of occupation type, jobs-housing balance of self-employed individuals is better than that of people who work for government agencies and public institutions, which is consistent with what we know. Self-employed individuals have more freedom in choosing workplace and higher income; thus, it is easier to achieve jobs-housing balance. Moreover, Kwon [56] also believed that occupation type is a crucial factor affecting commuting patterns. His research showed that since the service industry is more evenly distributed throughout the region, people working in the service industry may shorten their commuting distance. This is consistent with the conclusion of this paper because most self-employed individuals work in the service industry in Zoucheng County.

Subsequently, from the household level, only the household structure has a significant impact on the jobs-housing balance. The possible reason is that families with different family structures have different priorities. For example, single families pay more attention to suitable living conditions, nuclear families are mostly centered on the children, and the stem families have to take care of the elderly except for the children. However, all these families' common purpose is to maximize the family interests. Kolodin Ferrari [57] verified the relationship between household structure and jobs-housing balance using the accessibility index. It was found that household size is one of the significant factors, that is, the more household members, the lower the accessibility level, which fully verifies the conclusion of this paper.

Last noted but equally significant, the built environment has the most important influence on the jobs-housing balance. In this level, eight factors are selected, and four factors that passed the significance test are the residential location, commuting way, walking time to the nearest bus stop, and housing tenure. From the residential location, the jobs-housing balance of residents living in towns is the best. The possible explanation is that Zoucheng County has a preferable level of economic development in town, which can provide a large number of jobs for nearby residents, and the income level is considerable. From the perspective of commuting way, the jobs-housing balance of people who take private transportation is better than that take public traffic. Through the survey data, it was found that only 50% of the people can reach the bus stop within 5 min, which indicates that the bus system in Zoucheng County is underdeveloped and walking time to the bus stop is long, which leads to a low degree of jobs-housing balance among people who take public traffic. However, people who have a high income usually choose private cars for commuting, significantly shortening commuting time compared with public transportation, and consequently, they have a sound jobs-housing balance. In the study, Bwire [58] believed that people who take public transport have a high degree of jobs-housing balance, which is different from the conclusion of the paper. It may be due to different research areas and various stages of urbanization, resulting in distinct development levels of public transport systems in cities. Studies have shown that when a city's public transportation system is developed, people are more willing to commute by public transportation [59]. Because compared with private transportation, the commuting cost of public transportation is low, and the commuting time will be shortened (Buses generally have dedicated lanes and will not in traffic congestion) The walking time to the nearest bus stop is a significant factor affecting the jobs-housing balance, which is similar to the conclusion drawn by Guo [60]. In this study, the relationship between walking time to the nearest bus stop and the jobs-housing balance is that the possibility of individuals achieving the jobs-housing balance first decreases and then increases with the increasing walking time to nearest the bus stop. The reason for this phenomenon may be that a farther distance from the nearest bus stop leads to longer commuting distance and time, higher commuting cost, and poorer degree

of jobs-housing balance. Moreover, when the distance between residential place and the nearest bus stop exceeds a certain distance and the marginal cost of taking a bus increases, people may choose private cars. The reason is that compared with the bus, driving a car can shorten the commute time, and the saved time can offset the cost of using a car, which is easier to achieve jobs-housing balance. From the perspective of housing tenure, the degree of jobs-housing balance of commuters whose house without property rights is higher than that of people with housing property rights, and the people without housing property rights here mainly refer to renters and people living in dormitories. In this case, people who do not own houses with property rights may have a lot of freedom to choose their residential place; thus, there is more potential to achieve the jobs-housing balance. Notably, it is a preliminary attempt in this field to explore the impact of housing property rights on jobs-housing balance.

*5.2. Factors of Various Dimensions with Different Influences on Residents' Jobs-Housing Balance*

After using binary logistic regression model to analyze the influencing factors of jobs-housing balance, the CART model was applied to score the intensity of the factors that passed the significance test. The results of the scoring are in descending order: occupation type, commuting way, residential location, walking time to the nearest bus stop, housing tenure, gender, and household structure. According to the score of each influencing factor, the influencing factors are divided into three categories in this paper, namely, key factors (score > 1), important factors (score between 0.5 and 1), and auxiliary factors (score < 0.5). Therefore, the key factors are occupation type, commuting way, and residential location; the important factors are walking time to the nearest bus stop and housing tenure; the auxiliary factors are gender and household structure. It can be found that one of the key factors is the individual dimension, two are built environment dimension; all the important factors are built environment dimension; and one of the auxiliary factors is the individual dimension, whereas the other is household dimension. Specifically, occupation type is the most important factor influencing the jobs-housing balance, and the possible explanation is that the occupation of an individual is generally related to the way it commutes, and then determines the income of the individual; thus, its impact on the balance of work and housing is the primary. In addition, commuting way and residential location are also the key factors affecting the jobs-housing balance, because commuting way indirectly determines the commuting time of individual, and the number and quality of jobs provided by different residential locations are distinct; thus, these two factors significantly affect the jobs-housing balance. Furthermore, from the point of view of important factors, walking time to the bus stop and housing tenures should be specifically focused on. Furthermore, gender is an auxiliary factor affecting the jobs-housing balance, which shows that the latter is almost independent of the former, which is also in line with common sense, as woman become more involved in economic activities, their social status and household status are increasing; thus, the degree of jobs-housing balance has been comparable to that of men. In the end, household structure is an auxiliary factor influencing the jobs-housing balance. This is inconsistent with our hypothesis that the family dimension is at the core that affects the individual's jobs-housing balance, which may be related to the quality of urbanization in Zoucheng County. Although the urbanization rate of Zoucheng County has reached 64.8%, it is still economic development-oriented urbanization that only emphasizes speed, rather than people-oriented high-quality urbanization that pursues the whole family to settle down in the city. Overall, the driving mechanism framework of influencing factors on jobs-housing balance of residents in peri-urbanization areas is constructed, as shown in Figure 11.

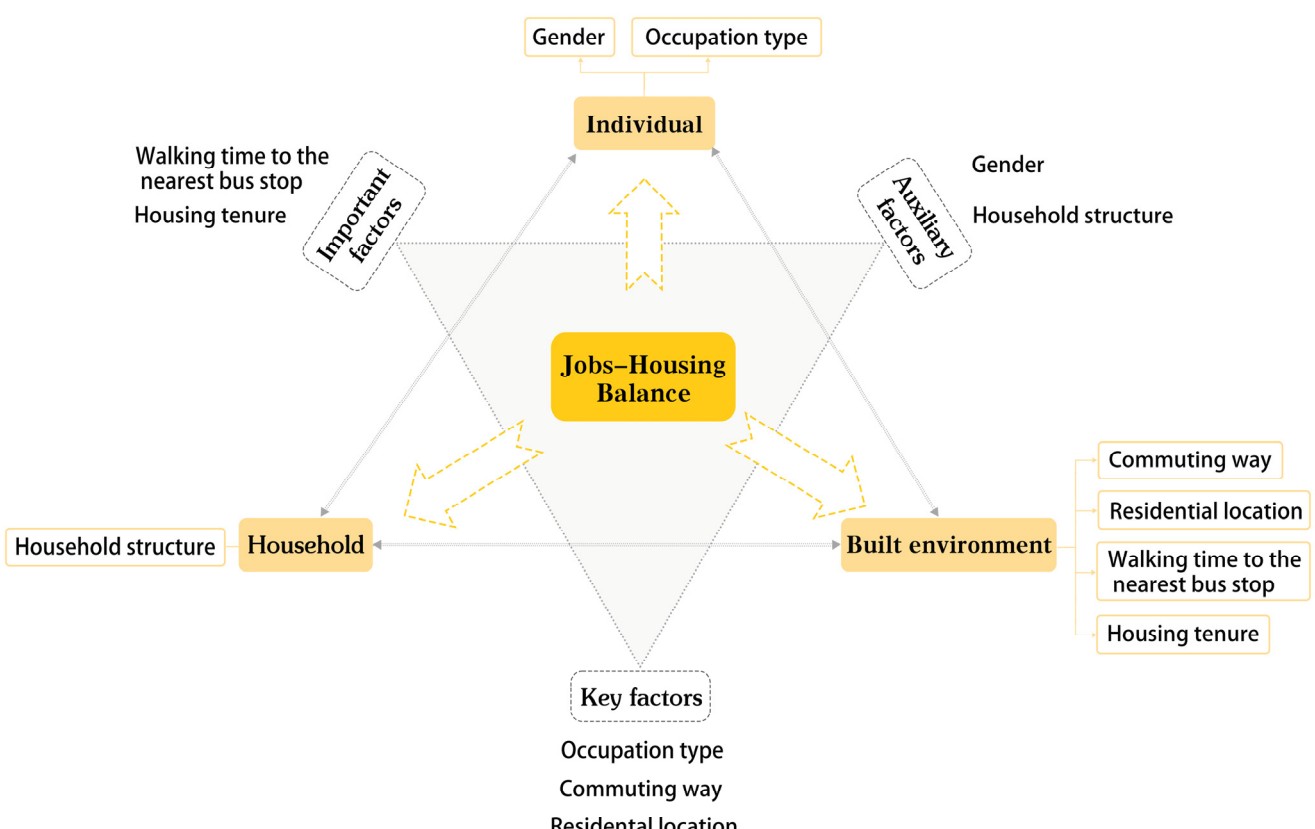

**Figure 11.** Influencing mechanism of jobs-housing balance.

### 5.3. Suggestions and Limitations

The results in this paper are of great significance to promoting the jobs-housing balance in county areas and realizing high-quality urbanization. This paper holds that jobs-housing balance is a complex spatial process, which is influenced by individuals, families, and the built environment, and the influence degree of each influencing factor is quite different. Through the empirical analysis, the following policy suggestions can be provided for city managers. First of all, it is necessary to consider the diversity of local residents' occupation types and provide them with jobs that match their skills. Secondly, due to its particularity, peri-urbanization areas should focus on the regulation of jobs-housing balance in urban areas and villages, including vigorously developing local public transport systems, improving public transport service levels, encouraging people to travel by public transport, and improving public transport facilities near residences. Finally, in addition, the government should vigorously develop the rental market, adhere to the principle and policy of both renting and purchasing houses, and provide suitable housing for staff near their workplace.

The theoretical value of this study mainly has the following three points: First, it enriches the theoretical connotation of the jobs-housing balance, as the previous research only used commuting cost to measure the jobs-housing balance, including commuting distance and commuting time. In this paper, however, the benefit index of jobs-housing balance is proposed, which involved commuting benefit on the basis of commuting cost. Secondly, it reveals that discussing the influence of household on the jobs-housing balance is not appropriate in China's counties at present. Finally, it provides a new research framework and method for follow-up scholars to further explore the county-level jobs-housing balance, which is also conductive to further study the driving mechanism of it. From a practical point of view, this study can provide a reference for government agencies to scientifically formulate relevant strategies, regulate residents' jobs-housing status to achieve the jobs-housing balance, and help counties achieve higher-quality urbanization.

However, there are also some limitations, including the following two points: First of all, with the emergence of big data, and with its characteristics of wide coverage, strong timeliness, etc., many scholars have carried out research on the visual expression and accurate analysis of the jobs-housing relationship based on bus cards, mobile phone signaling, and multi-source LBS data. Therefore, in the subsequent analysis of the influencing factors of jobs-housing balance, it is possible to consider combining traditional questionnaire data with big data for a more accurate analysis. In addition, based on the cost-to-income ratio in the financial fields, the benefit index of jobs-housing balance is a preliminary attempt to expand the connotation of jobs-housing balance theory. However, the selection of relevant indexes and the weight of each index still need to be scientifically defined, which is also the direction that following scholars should strive for. For instance, the housing cost is a key factor to determine location in the urban center, which should be taken into account when constructing the benefit index of jobs-housing balance. Finally, we sincerely call for more researchers to join us to provide a more accurate understanding of academia and society.

## 6. Conclusions

Taking Zoucheng County, China as an example, based on 1110 questionnaires of non-agricultural employees in Zoucheng County issued by the research group in 2021, this paper first constructed the benefit index of jobs-housing balance and then analyzed the influencing factors of jobs-housing balance of non-agricultural employees in Zoucheng County using binary logistic regression and the random forest classification and regression tree. Through empirical analysis, this paper draws the following conclusions: Firstly, 57% of the residents of Zoucheng County have achieved the jobs-housing; thus, the balance rate of jobs-housing in the county is in a better state compared with some international metropolises. Secondly, individual, household and built environment dimensions jointly affect jobs-housing balance of residents, and there are seven factors in total. From the perspective of individual, factors that passed the significance test are gender and occupation type. Moreover, household structure passed the test in the household dimension, and residential location, commuting way, walking time to the nearest bus stop, and housing tenure passed in the built environment dimension. As a result of that, household is not the core dimension that affects the jobs-housing balance at the current stage of China's county areas. Thirdly, factors have various influencing degrees on the residents' jobs-housing balance, and the order of these influencing factors is: occupation type > commuting way > residential location > walking time to the nearest bus stop > housing tenure > gender > household structure. According to the score of each factor, the influencing factors can be divided into three categories, which are the key factors, including occupation type, commuting way, and residential location; important factors, including walking time to the nearest bus stop and housing tenure; auxiliary factors, including gender and household structure. Therefore, urban managers should focus on the impact of occupation type, commuting way, and residential location on jobs-housing balance.

**Supplementary Materials:** The following supporting information can be downloaded at: https://www.mdpi.com/article/10.3390/su14137921/s1, Questionnaire.

**Author Contributions:** Conceptualization, H.Z. (Haonan Zhang) and H.Z. (Hu Zhao); methodology, H.Z. (Hu Zhao) and H.Z. (Haonan Zhang); software, S.M.; validation, Y.Z., S.M. and H.Z. (Haonan Zhang); formal analysis, H.Z. (Haonan Zhang); investigation, S.M.; resources, S.M.; data curation, Y.Z.; writing—original draft preparation, H.Z. (Haonan Zhang); writing—review and editing, H.Z. (Haonan Zhang); visualization, Y.Z.; supervision, H.Z. (Hu Zhao); project administration, H.Z. (Hu Zhao); funding acquisition, H.Z. (Hu Zhao). All authors have read and agreed to the published version of the manuscript.

**Funding:** This research was funded by the National Natural Science Foundation of China (Grant No.51878393).

**Institutional Review Board Statement:** Not applicable.

**Informed Consent Statement:** Not applicable.

**Data Availability Statement:** Not applicable.

**Acknowledgments:** First of all, I would like to express my gratitude to reviewers and editors for your industrious payout, as you reviewed our paper and put forward such constructive suggestions. Secondly, special thanks are due to my colleagues Wang Xiaotong who had made this paper possible through significant support.

**Conflicts of Interest:** The authors declare no conflict of interest.

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
