# Peer review of "Research on the Jobs-Housing Balance of Residents in Peri-Urbanization Areas in China: A Case Study of Zoucheng County"

_sustainability, doi:10.3390/su14137921_

Round 1

Reviewer 1 Report

The manuscript focuses on the job-housing balance in peri-urbanization areas in China. The authors propose a benefit index of the job-housing balance and select the main factors influencing this balance by applying logistic regression, random forest classification and regression tree to a database of 1110 valid samples. The work is appreciable and interesting. There are, however, some methodological inconsistencies related to the cost-benefit analysis.

Lines 370-374 - The CBA is an analysis of alternative investments in monetary terms and refers to flows of costs and benefits over time expressed in terms of net present value. In the manuscript there are no alternatives, no value flows over time, and no net present value calculations. Although the authors relate costs and benefits, this calculation cannot be considered an application of cost-benefit analysis. This claim is scientifically flawed.

Moreover, the proposed index is not particularly innovative and, indeed, does not take into account the cost of housing, which is a key element in determining location. Studies in urban economics conducted as early as the 1960s, for example by William Alonso, addressed the issue of location choice in much more detail.

In any case, both in the first sections (lines 108, 263) and in the discussion section (lines 712-713), the index's reference to CBA theory must be removed, simply because it has not been applied.

Comparing benefits with costs does not mean that a cost-benefit analysis is being done.

Other suggestions

Lines 359-362. Results are presented before even explaining which method will be applied. Present the results after the method.

Line 310 - It is suggested that the questionnaire be presented in an appendix to the paper.

Line 345 - The authors could explain what hukoustatus is.

Line 602 - It is suggested to replace “housing type” (which is very general) with “housing tenure”, since the only housing characteristic considered is house with/without property rights.

Author Response

Dear reviewer:

Thank you for your busy schedule can find time to review our manuscript entitled “Research on the Jobs-Housing Balance of Residents in Peri-Urbanization Areas in China: A Case Study of Zoucheng County” (ID: sustainability-1760690). Those comments are all valuable and very helpful for revising and improving our paper, as well as the important guiding significance to our research.

We have studied the comments carefully and have made corrections which we hope meet with approval. For your convenience, we will highlight the revised parts in yellow.

Finally, on behalf of all my partners, I’d like to thank you again for your review. We sincerely wish you success in your work and good health. See the attachment for detailed solutions.

Sincerely,

Mr.Haonan Zhang, Hu Zhao, Yanghau Zhang and Ms. Saisai Meng

School of Architecture and Urban Planning, Shandong Jianzhu University

[email protected]

Reviewer 2 Report

Overall, I think this is a sound paper structured and written in an engaging way, clear about its purpose and firm in its academic underpinnings. I don't have much to say about the statistical techniques used in this paper - this area is outside of my expertise. 

The text needs to be read-through thoroughly  - there are quite a lot of typographical errors and sections where the meaning is not entirely clear. 

I note that the questionnaire data was collected in November 2021 - do you believe that the travel patterns reported were affected by the Covid pandemic? You should have a statement about this. 

It would be useful to know a bit more about the sectors / companies in which workers are being employed in Zoucheng. Isn't a key aspect of the job-housing balance a match between worker skills and employment offered? E.g. if much of the employment is unskilled or semi-skilled (e.g. manufacturing) then it offers a much greater possibility of workforce demand being met locally by residents with modest education levels. 

To continue, do ensure that you maintain precision and support for all of your claims made, especially in sections 5 and 6 where some of the assumptions expressed appeared to be rather sweeping - e.g. in 5.1 what does the literature say about when people switch from public transport to private cars? (noting that private cars can be problematic for their users in various respects). 

Author Response

Dear reviewer:

Thank you for your busy schedule can find time to review our manuscript entitled “Research on the Jobs-Housing Balance of Residents in Peri-Urbanization Areas in China: A Case Study of Zoucheng County” (ID: sustainability-1760690). I have to say that your suggestions are the beacon navigating for us to revise our manuscript to result in a better one.

We have studied the comments carefully and have made corrections which we hope meet with approval. For your convenience, we will highlight the revised parts in yellow.

Finally, on behalf of all my partners, I’d like to thank you again for your review. We sincerely wish you success in your work and good health. See the attachment for detailed solutions.

Sincerely,

Mr.Haonan Zhang, Hu Zhao, Yanghau Zhang and Ms. Saisai Meng

School of Architecture and Urban Planning, Shandong Jianzhu University

[email protected]

Reviewer 3 Report

The present article is focused on analyzed peri-urbanization process and the impact on jobs-housing balance of residents in peri-urbanization areas in China.

Abstract is overall well developed with the mention that it is not clearly mentioned the conclusion of this work.

Introduction is not sufficiently sustained by international actual references. The focus is only on references from China. In this situation, which is the international relevance of your study considering that peri-urbanization is a worldwide process?

Considering the characteristics of studied area, the number of applied questionnaires is relatively low. 

Figure 11 should be improved in terms of quality.

In the abstract is mentioned that relevant suggestions to promote jobs-housing balance of residents in county area were put forward. However, this part is clearly underdeveloped. These suggestions are not sufficiently developed and correlated with the results of your study. 

Also, the questionnaires content is insufficiently described being rather difficult to analyze its relevance.

Author Response

Dear reviewer:

Thank you for your busy schedule can find time to review our manuscript entitled “Research on the Jobs-Housing Balance of Residents in Peri-Urbanization Areas in China: A Case Study of Zoucheng County” (ID: sustainability-1760690). Your suggestions are absolute of great importance and have fundamentally improved the quality and readability of this paper.

We have studied the comments carefully and have made corrections which we hope meet with approval. For your convenience, we will highlight the revised parts in yellow.

Finally, on behalf of all my partners, I’d like to thank you again for your review. We sincerely wish you success in your work and good health. See the attachment for detailed solutions.

Sincerely,

Mr.Haonan Zhang, Hu Zhao, Yanghau Zhang and Ms. Saisai Men

School of Architecture and Urban Planning, Shandong Jianzhu University

[email protected]

Round 2

Reviewer 1 Report

The authors made all the changes or additions requested.

In my opinion, the article is suitable for publication.